# Full-field structured-illumination super-resolution X-ray transmission microscopy

Benedikt Günther[1,2,3], Lorenz Hehn[1,2,4], Christoph Jud[1,2], Alexander Hipp[5], Martin Dierolf [1,2] & Franz Pfeiffer[1,2,4]

Modern transmission X-ray microscopy techniques provide very high resolution at low and medium X-ray energies, but suffer from a limited field-of-view. If sub-micrometre resolution is desired, their field-of-view is typically limited to less than one millimetre. Although the field-of-view increases through combining multiple images from adjacent regions of the specimen, so does the required data acquisition time. Here, we present a method for fast full-field super-resolution transmission microscopy by structured illumination of the specimen. This technique is well-suited even for hard X-ray energies above 30 keV, where efficient optics are hard to obtain. Accordingly, investigation of optically thick specimen becomes possible with our method combining a wide field-of-view spanning multiple millimetres, or even centimetres, with sub-micron resolution and hard X-ray energies.

[1] Department of Physics, Technical University of Munich, James-Franck-Str. 1, 85748 Garching, Germany. [2] Munich School of BioEngineering, Technical University of Munich, Boltzmannstr. 11, 85748 Garching, Germany. [3] Max-Planck-Institute of Quantum Optics, Hans-Kopfermann-Str. 1, 85748 Garching, Germany. [4] Department of Diagnostic and Interventional Radiology, Technical University of Munich, Ismaninger Str. 22, 81675 Munich, Germany. [5] Helmholtz-Zentrum Geesthacht, Max-Planck-Str. 1, 21502 Geesthacht, Germany. Correspondence and requests for materials should be addressed to B.G. (email: benedikt.guenther@tum.de)

X-ray transmission microscopy is a standard method for non-destructive testing at low and medium X-ray energies providing valuable insights into a specimens' microscopic structure due to the high resolution achievable. They can be classified into three categories: scanning transmission X-ray microscopy (STXM), full-field transmission X-ray microscopes (TXM), and microscopes based on geometric magnification in a cone beam geometry[1,2]. In STXM, the X-ray beam is focused onto the specimen and the transmitted intensity at that particular point is recorded. Subsequently, the object is scanned through the focus whose size determines the resolution of this technique[3]. In contrast to this procedure, in full-field TXM a condenser optic creates an extended illumination on the object, which in turn is re-imaged onto a two-dimensional detector by magnifying optics[4]. If sub-micrometre resolution is to be achieved, the field-of-view of all these techniques is typically limited to <1 mm[5]. While the field-of-view increases when combining multiple images from adjacent regions of the specimen, so does the required data acquisition time.

Here we show a technique that is conceptually different to above-mentioned state-of-the-art approaches that rely on a iso-lated single X-ray beam: Instead of using a single isolated pencil beam, creating a whole periodic array of such beams allows for simultaneous illumination of larger parts of the specimen also in a STXM, similar in concept to multi-beam scanning electron microscopy[6,7]. The signals from the multiple beams have to be recorded with a two-dimensional detector on which the con-tributions of the individual beams have to be separable. A com-plete image is obtained by scanning the specimen over one period of the array of illuminating beamlets. Therefore, acquisition time can be drastically reduced if these beamlets are spaced closely, i.e. on the order of micrometres. Such a periodic X-ray intensity modulation is created by the Talbot effect[8,9]. It is the occurrence of self-images of a periodic structure at distinct distances along the optical axis when illuminated with a spatially coherent wave-field. In the X-ray regime, this effect has been mainly exploited for interferometric phase-contrast and dark-field imaging[10–13]. Self-images of typically employed binary gratings with periods on the order of a couple of micrometres exhibit roughly the same shape as the original grating structure. While this is not sufficient for sub-micrometre resolution, non-binary gratings change the longitudinal as well as the transverse intensity profile of the self-images[14]. The desired structured illumination with an array of narrow peaks is generated selecting an appropriate non-binary grating structure.

## Results

### Simulation of structured full-field illumination.
In general, tilting a binary grating results in an effective trapezoidal height profile (Supplementary Fig. 4). At certain angles, this becomes triangular, which produces the sharpest intensity peaks. Numer-ical free-space propagation of the wave-front exiting a simulated triangular grating yields the Talbot-carpet in Fig. 1. For the highest resolution, the specimen should be placed in the trans-verse plane with minimum peak width. This plane can be extracted from Fig. 1b. The intensity profile in this plane is depicted in Fig. 1c and features main peaks with a full-width at half-maximum (FWHM) width of 0.41 µm. The corresponding second-moment width of the total intensity distribution is 0.73 µm (r.m.s., here and in the following used as the sigma value of a corresponding Gaussian distribution), even at X-ray energies as high as 35 keV. This corresponds to a full width of the central peak where the intensity dropped down to ~10% of the peak intensity. Thereby, triangular gratings provide a means to create a sub-micrometre structured illumination whose resolution is

entirely independent of detector pixel size as long as the latter is smaller than the grating period. This separation of resolution and detector pixel size enables super-resolution imaging.

### Demonstration of structured full-field illumination.
As a first experimental step, we measured the Talbot-carpet of the grating with an effective triangular profile. The experiments were per-formed at an X-ray energy of 35 keV at beamline P05 at PETRA III at DESY (Hamburg, Germany). A schematic of the experi-mental set-up used for all experiments is depicted in Fig. 2. The result of the Talbot-carpet measurement is shown in Fig. 3. The measured intensity modulation in propagation direction agrees very well with the simulated one (c.f. Fig. 1). The slight drift of the periodic illumination along the vertical axis of the image is attributed to the limited positioning accuracy of the detector. The modulation strength of the transverse intensity profiles is char-acterized through their variance (Fig. 3b): the higher the variance, the stronger the intensity changes. The longitudinal intensity modulation agrees well with the predicted one. The measured modulation depths of the transverse profiles in Fig. 3d–f are much smaller than the ones simulated for an ideal situation, c.f. Fig. 1d–f. This is the result of mainly two contributions: First, elements stabilizing the grating lines, so-called bridges[15,16], dis-cussed in detail in the "Methods" section with a sketch of the grating structure depicted in Supplementary Fig. 4, introduce a homogeneous background signal that reduces the modulation depth when averaging over the detector lines. Second, source blur and, much more importantly, intrinsic detector point-spread function (PSF) of about 1.3–1.5 µm (r.m.s.) (Supplementary Note 1 and Supplementary Fig. 1) significantly reduce the detectable intensity modulation. The effect of the source blur and bridges on the Talbot-carpet is shown in Supplementary Fig. 2. At the position of our structured illumination (Supplementary Fig. 2c), the source blur of ~27 nm is very small compared to the size of the individual illuminations of ~0.7 µm and thus barely affects the intensity modulation. If the detector resolution is included into the simulation additionally, the modulation depth apparent on the detector is reduced to 30% and the intensity enhancement to <1.1 (Supplementary Fig. 3). This is in good agreement with the measured modulation demonstrating that the measured modulation is dominated by the detector PSF. Slightly lower signal modulation in the measured data is attributed to imperfections arising from vibrations of the monochromator and the finite coherence length. Consequently, the intensity modula-tion in the focus is close to the ideal case, depicted in Fig. 1c.

### Demonstration of super-resolution imaging.
For a line grating, the created Talbot-carpet is translationally invariant along the direction of the grating lines. As a result, super-resolution is generated only in one dimension, perpendicular to the grating lines, which is, in our case, the horizontal direction. In the vertical direction, parallel to the grating lines, the resolution is limited by the detector. This enables direct comparison of the intrinsic detector resolution with the achieved super-resolution and therefore is highly suited for demonstration purposes.

In order to quantify the resolution, a lithographically produced resolution test chart (Xradia, Pleasanton, USA) was placed in the first focal plane. A detailed description of the experiment and the signal extraction is given in the "Methods" section. The reconstructed resolution test chart is depicted in Fig. 4a. The vertical lines in the inset appear much sharper than the horizontal ones demonstrating the increased resolution in horizontal direction qualitatively. A zoom on horizontal and vertical lines with a width of 1 µm is displayed as an inset in Fig. 4. It qualitatively shows the gain in resolution in

**Fig. 1** Simulated Talbot-carpet of a triangular grating at an X-ray energy of 35 keV. The grating's period is set to 5 μm and its height to 32 μm. **a** One full Talbot-distance of the Talbot-carpet is depicted for two simulated grating periods. **b** Variance perpendicular to the propagation direction. The distances for structured illumination of the object are extracted from this graph. **c** The transverse intensity profile at the resulting specimen position. **d**–**f** show transverse intensity profiles at certain locations within the Talbot-carpet, which are indicated by lines in the corresponding colour and Roman numerals in **a**

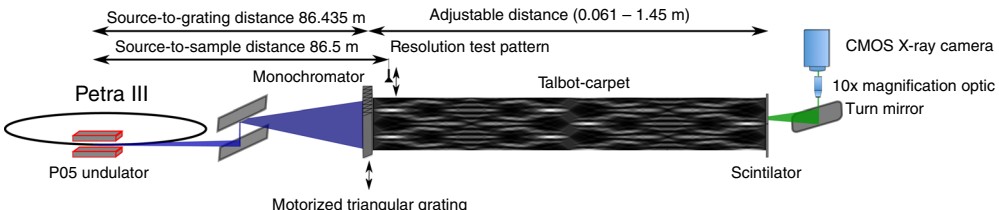

**Fig. 2** Schematic of the experimental set-up used. X-rays emerging from an undulator are monochromatized and propagating to the triangularly shaped grating, which can be moved in the horizontal direction with a piezo-electric actuator. At a distance of 65 mm downstream the grating, the specimen (resolution test chart) is placed on a sample positioning stage. Another 5 mm away, the detector is placed. For the Talbot-carpet measurement, the grating to detector distance first is slightly reduced and then the detector is moved in steps of 1 mm away from the grating, up to a maximum grating to detector distance of 1.43 m

the horizontal direction as the vertical lines appear much sharper than the horizontal ones. In order to quantify this effect, we pursued two approaches. First, we determined the modulation transfer function (MTF) from the line pair patterns marked in Fig. 4a, independently for the horizontal (orange box) and vertical (green box) direction. The raw data averaged along the short axis of the marked regions is depicted in Fig. 4c.

This data is used to calculate the MTF shown in Fig. 4d and referenced to the amplitude for the line pair with the largest width. In contrast to the MTF in the vertical direction, which decreases with increasing spatial frequency, the MTF in the horizontal direction exhibits a strong peak around 200 lp mm$^{-1}$. This corresponds to the spatial frequency of the structured illumination (5 μm) where we would expect increased sensitivity. In general, the horizontal MTF is higher than the vertical one and

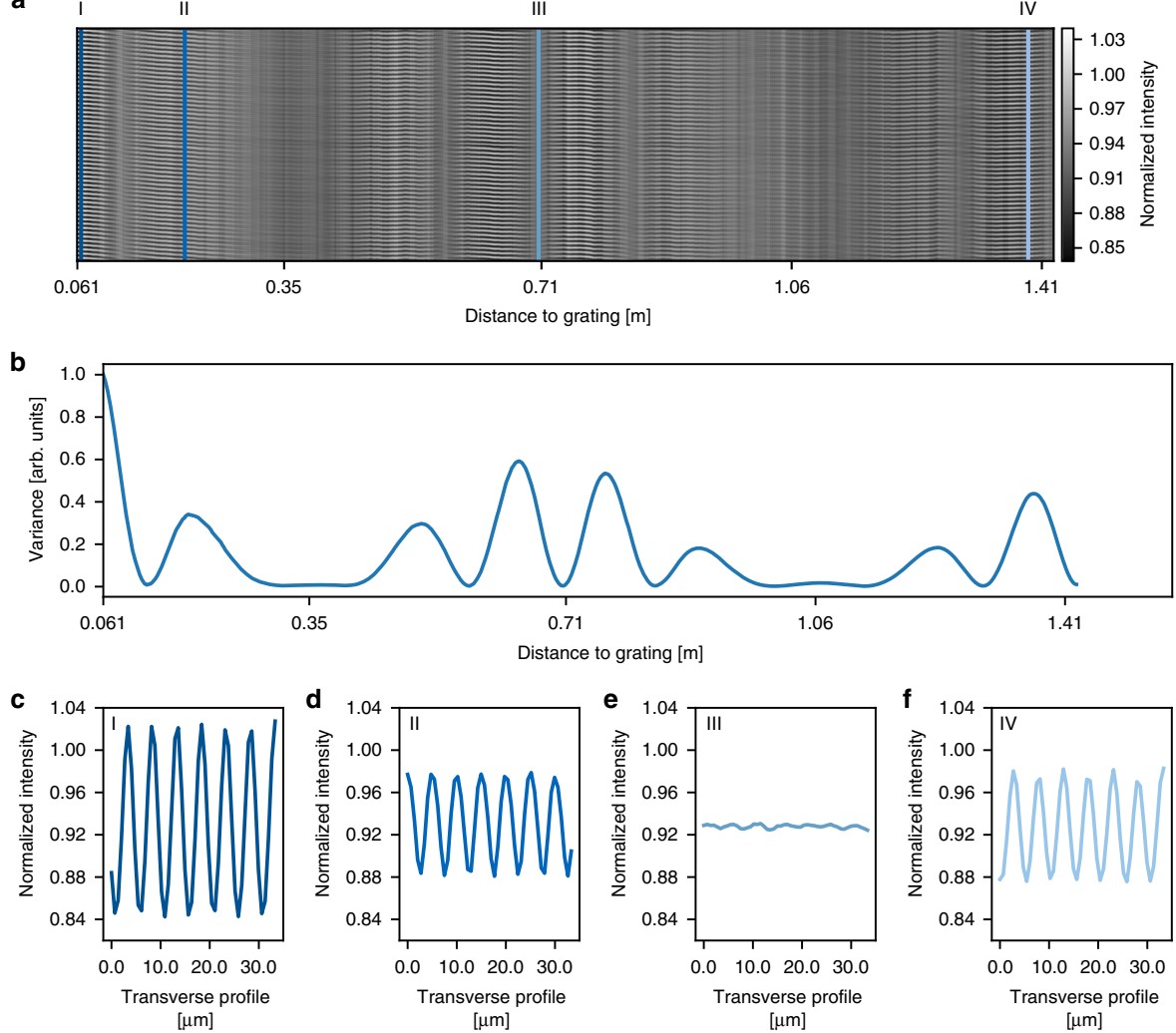

**Fig. 3** Talbot-carpet of a triangular grating measured at an X-ray energy of 35 keV. The grating's period is 5 μm and its height is 32 μm. **a** Measured Talbot-carpet, from 61 mm behind the grating up to one full Talbot-distance. **b** Variance of a transverse intensity profile as a function of distance behind the grating. **c** Transverse intensity profiles through the Talbot-carpet at the specimen location (at the same position as in Fig. 1). **d–f** Transverse intensity profiles at the same positions as in Fig. 1d–f, which are also indicated by lines in the corresponding colour and Roman numerals in **a**

not even approaching 10% at 500 lp mm$^{-1}$, while the 10% cutoff in the vertical direction is at ~350 lp mm$^{-1}$, which corresponds to a half-period of ~1.5 μm. Therefore, we calculated the edge-spread function in both directions in order to determine the absolute resolution. While the vertical half-period resolution determined with this technique is 1.49 μm, which agrees well with the intrinsic detector resolution determined from a reference image in standard parallel beam imaging geometry (Supplementary Fig. 1), the horizontal resolution of 0.59 μm is more than twice as good. The latter corresponds to a full-period resolution of 1.18 μm, which is 1.6 times the width of the individual beamlets in the structured illumination. In conclusion, this resolution well below the intrinsic limit of the detector itself demonstrates full-field super-resolution microscopy that is entirely limited by the width of the individual beamlets of the structured illumination.

## Discussion

In this experiment, we intentionally used a one-dimensional grating for demonstration purposes. Implementation of a two-dimensional grating, successfully applied to Talbot interferometry initially[17,18], would provide super-resolution in both dimensions.

Further optimization of the grating shape is foreseen reducing the beamlet width and therefore improving resolution[19,20].

The field-of-view available in the experiment enabled us to scan an effective number of 2200 resolution elements s$^{-1}$. At an—for the grating used here—optimum detector pixel size of 5 μm, the field-of-view of a detector with the same number of pixels would allow to scan 16,430 resolution elements s$^{-1}$ at a field-of-view of 26.5 mm. Guizar-Sicairos et al. acquire 25,000 resolution elements s$^{-1}$ with their high-speed set-up at an X-ray energy of 6.2 keV[21]. Scaling acquisition time with the relative efficiency of their Eiger detector at 35 keV, their acquisition speed would be limited to about 2500 resolution elements s$^{-1}$ at this energy. This is close to the speed we achieved already in our proof-of-principle experiment and far below the speed of an optimized set-up.

In Supplementary Note 2, we present a detailed comparison with a hypothetical pencil-beam scanning microscope generating the same focal spot size of 0.7 μm (r.m.s. determined with second moment) and operating at the same X-ray flux. In order to cover the same field-of-view as our approach, an orders-of-magnitude-larger number of lines has to be scanned with such a system. The acceleration of state-of-the-art linear stages thus becomes a strong limitation. As a consequence, our method is faster by more than

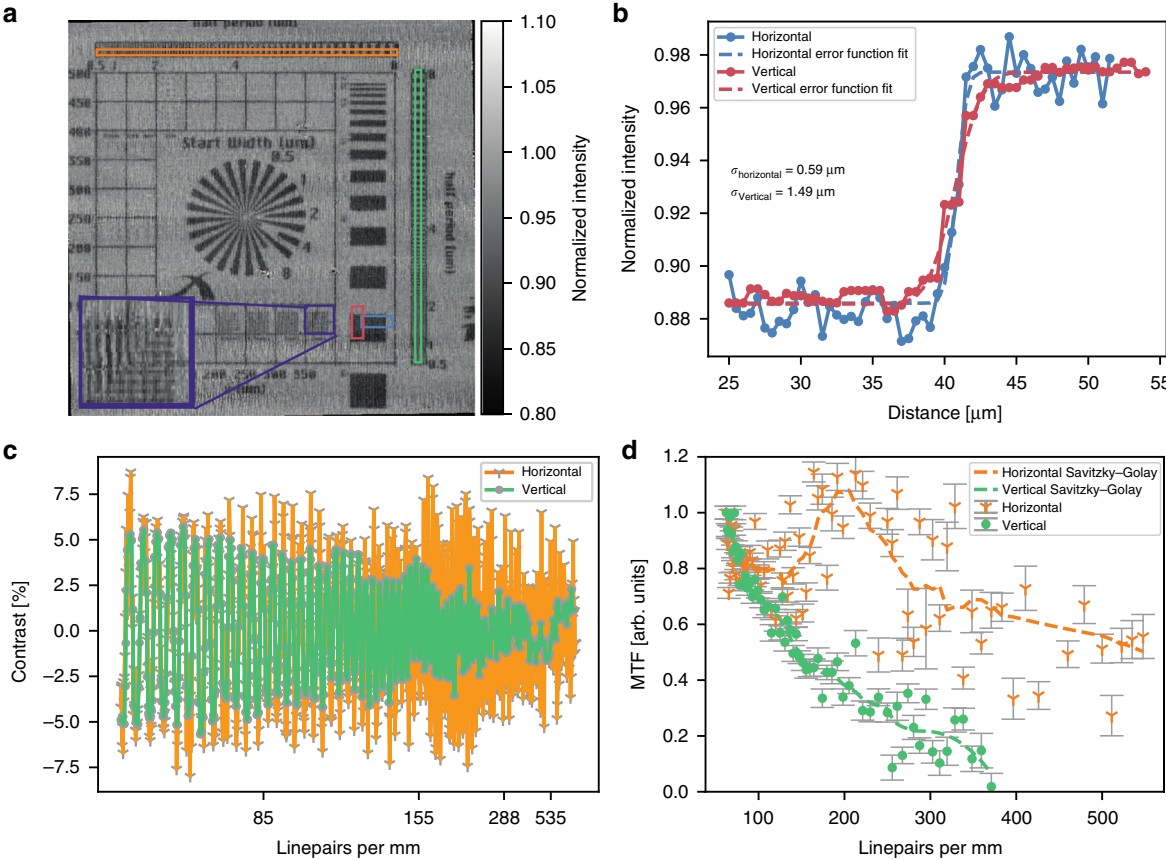

**Fig. 4** Analysis of the achieved super resolution. **a** The reconstructed resolution test chart. The inset is a zoom-in on the lines framed by a purple box in the main image. The vertical lines in the inset are much sharper than the horizontal ones, illustrating the expected resolution gain in the horizontal direction. **b**–**d** The results of two quantitative analyses. **b** The edge-spread function that has been calculated for both directions, the horizontal one in blue and the vertical one in red, exhibits a significant gain in resolution for the horizontal edge. Displayed is only the region around the edge. **c** Raw data for the modulation transfer function (MTF) analysis. The horizontal direction is depicted in orange and the vertical one in green. **d** The resulting MTF values demonstrate a similar performance gain. The dashed lines represent the MTFs after application of a Savitzky–Golay filter of third order with a width of 21 data points. The increased MTF around 200 lp mm$^{-1}$ is attributed to the fact that this is the spatial frequency of the structured illumination. Error bars indicate the statistical uncertainty. Vibrations of the monochromator cause artifacts in the image that translate into fluctuations of the MTF that cannot be quantified. The regions of interests for both analyses are indicated as boxes of the corresponding colour in **a**

one order of magnitude for typical scan parameters. In addition, Supplementary Note 2 also includes a comparison with a conventional transmission X-ray microscope.

In a conventional parallel beam geometry, the detector pixel size has to be at least $500 \times 500$ nm, which can be reached with an optical microscope magnifying the scintillator plane onto the detector. In contrast, in our approach the flux of an area of $5 \times 5$ μm, so 100 times larger than in the parallel beam case, is focused with an efficiency of 52.8%. Compared to the unfocused beam in the parallel beam microscope, the X-ray flux in the focus is thus enhanced by a factor 52.8. In an optimized set-up for our multibeam approach, the signal from one focus is then recorded in one detector pixel. Data acquisition with our proposed scanning approach requires $10 \times 10$ steps to cover the whole field-of-view. Considering the flux enhancement factor of 52.8, the exposure time per image can be reduced accordingly, which results in roughly twice the acquisition time as in the conventional parallel beam geometry for a full image. However, in one such $10 \times 10$ scan, 100 times the field-of-view of a single parallel beam image is covered by the multi-beam approach. To reach the same field-of-view in the parallel beam geometry, 100 images have to be taken, too. Therefore, our proposed method is >50 times faster than stitching frames from conventional parallel beam imaging with the same resolution, which requires the same number of

acquisitions and thus motor movements generating the same amount of data and a similar scanning overhead. Furthermore, the required optical microscope imaging the scintillator plane onto the detector has typically a low efficiency. Another factor of 2 in efficiency thus could be gained using a fibre-coupled scintillator for the 5 μm pixels in the structured illumination case, which is not possible for the 0.5 μm effective pixel size provided by the ×10 optical magnification.

As large X-ray gratings with a diameter around 10 cm are readily available without stitching[22], the available field-of-view and thus the gain in acquisition time or resolution elements, respectively, is mostly limited by the extension of the X-ray beam and the number of pixels of the detector.

Moreover, we demonstrated focusing of hard X-rays—even well above 30 keV—down to focal spot sizes of a few hundred nanometres with our method in a very simple and compact set-up. This is quite remarkable for a sub-micron scanning approach as other focusing devices like lenses or Fresnel zone plates suffer from a low efficiency in this energy range. For compound refractive lenses, e.g. <10% of the incident flux is concentrated into the focus at 25 keV[23]. In contrast, the phase-shifting grating employed here makes 52.8% available within the focus width of 0.7 μm. This efficiency is calculated from the simulation considering the absorption of the nickel structures with an average

height of 16 μm. Although Kirkpatrick–Baez mirrors[24] provide a high reflectivity in the hard X-ray energy range and sub-micron resolution[25,26], our approach is advantageous due to the high number of parallel illuminations, as discussed above.

In conclusion, (hard) X-ray nano-tomography of specimens with an extent of several millimetres at a resolution of several hundred nanometres becomes feasible. Accordingly, for now we see our full-field structured-illumination super-resolution X-ray transmission microscopy approach as a complementary technique for larger specimens to the sub-50 nm STXM and TXM microscopes, which are limited to very small specimen sizes of the order of a few hundred micrometres. Combining our method with recently developed brilliant compact inverse Compton sources, whose coherence has been demonstrated to be sufficient for Talbot interferometry[27–29], is straightforward. Implementation of a source grating, known from Talbot–Lau interferometry[12], will make this technique feasible also at high power (rotating anode) X-ray tubes. Thereby significantly decreasing data acquisition time while enlarging the field-of-view compared to current systems usually based on microfocus tubes. Accordingly, our technique paves the way to high-speed sub-micrometre imaging and computed tomography of large specimen in a laboratory environment.

## Methods

**Triangular-shaped grating**. Experiments were conducted using a nickel grating with a period of 5 μm, a duty cycle of 0.5 and a height of 32 μm. It was produced by the Karlsruhe Institute of Technology (KIT, Karlsruhe, Germany). In order to produce an effective triangular height profile, lithographic exposure was performed at an angle of 4.5° with respect the surface normal resulting in tilted grating lines. These grating lines are interconnected by periodically arranged small structures, so called bridges, in order to stabilize the tilted grating lines. Two grating lines are connected with bridges spaced by a constant distance, while the bridges in the respective neighbouring gaps are offset by half this distance. This generates another periodicity with twice the grating period at the position of the bridges. A sketch of the grating structure is depicted in Supplementary Fig. 4.

**Simulation of the Talbot-carpet**. Five grating periods have been included in the simulation and the lateral step size along the triangular height profile is 0.01 μm. In addition, only the phase shift of the nickel grating is included into the calculation, and absorption within the grating is neglected. The wave-field impinging on the grating is assumed to be a coherent monochromatic plane wave with an energy of 35 keV with an amplitude set to unity. This is a good approximation to the experimental situation because the source-to-grating distance in the experiment is >86 m and the X-ray source is an undulator providing a very narrow divergence angle rendering the paraxial approximation valid. The propagated wave-field is calculated employing the Fresnel approximation for near-field diffraction[30,31]. The interference pattern was simulated up to a propagation distance of one Talbot-distance, which is 1.411 m at an X-ray energy of 35 keV with 2823 equidistant propagation steps. This results in a resolution of 0.5 mm in propagation direction. Additional simulations including the effects of absorption, source blur and detector resolution are displayed in Supplementary Figs. 2 and 3. The width of the peaks is calculated as the square root of the second moment of the transverse intensity distribution.

**Experimental set-up**. All experiments were performed at an X-ray energy of 35 keV at the micro-tomography end-station at the HZG beamline P05 at PETRA III at DESY in May 2017. This energy was chosen in order to demonstrate that our proposed method is compatible with hard X-rays. The X-ray source size is 36 × 6 μm with a divergence of 28 × 4 μrad at 10 keV[32]. As the X-ray source parameters are barely varying with X-ray energy, these numbers are also valid at 35 keV. X-rays were monochromatized by a double crystal monochromator. The FWHM horizontal extension of the X-ray beam at the sample position of the micro-tomography end-station (source-to-sample distance 86.5 m) is 5.6 mm[32]. The sample-to-detector distance can be adjusted up to 1.4 m. The detector was a CMOS camera developed by the Karlsruhe Institute of Technology and based on the CMOSIS CMV20000 chip. A ×10 optical magnification of the scintillator, lutetium–aluminum garnet with a thickness of 50 μm, onto the detector resulted in an effective pixel size of 0.68 μm. All experiments were performed using the same magnification, but 2 × 2 software binning of the raw data was performed for the microscopic experiment increasing the pixel size above the width of the individual illuminations produced by the Talbot-carpet.

**Talbot-carpet measurement**. Geometrical constraints of the set-up allowed a minimum sample-to-detector distance of 61 mm. The Talbot-carpet was measured in steps of 1 mm up to a propagation distance of 1.425 m. Acquisition time for each frame was 250 ms and 5 images were acquired at each distance. Only the last three images were averaged afterwards as the first two images were severely distorted by residual vibrations of the detector stage after movement. We performed two scans, one reference scan without the grating in the beam and one with the grating. Dark frames were acquired, averaged and subtracted from the individual averaged frames of both scans. The final image was obtained by dividing the corresponding frames of the grating scan by the ones of the reference scan. Although this usually eliminates intensity variations in the X-ray beam very well if the delivered X-ray beam is stable, vibrations of the monochromator resulted in some remaining background inhomogeneities in our case. In addition, limited positioning accuracy of the detector resulted in slight horizontal and vertical drifts of the Talbot-carpet over more than one grating period and bridge period, respectively. If the same detector line is tracked over the whole distance, such vertical drifts imply that the Talbot-image of the bridge-like structure is visible at some point instead of the one of the gratings. Bridges, as mentioned before, exist between all grating lines but are shifted in their position between two adjacent ones. Averaging over all detector rows therefore eliminates these undesired contributions at the cost of a slightly increased background reducing the depth of the intensity modulation.

**Resolution test chart measurement**. The resolution test chart was placed at a grating-to-sample distance of 65 mm, where the focus of the grating was located. The sample-to-detector distance was 5.5 mm, which was still sufficiently close so that a negligible change in width and intensity of the individual illuminations occurs. In order to demonstrate that the resolution is only limited by the width of the individual beamlets, 2 × 2 software binning was applied to the data creating an effective pixel size of 1.36 μm, which is well above the beamlets' size. In our case, the grating or, in other words, the structured illumination was scanned relative to the specimen by a piezo-electric actuator because it provided higher accuracy than the stages for the sample. We used 10 steps to cover one grating period of 5 μm. The acquisition time was 300 ms for each frame. A reference scan without the resolution test chart was acquired as well as dark frames.

**Signal extraction**. First, both, the reference frames and the sample frames are corrected subtracting the dark frame. Hot pixels and dead pixels are corrected in a second step. Therefore, as an intermediate step, the median of the frame with a size of 3 × 3 pixels is calculated and subtracted from the original one. If the absolute value of the difference was >120 counts, the value of this pixel was replaced by its median. As a next step, the 2 × 2 software binning of the images mentioned above was performed. The actual signal extraction works as follows: First, we identified the minima of the structured illumination within each row of the detector. For a region between two minima, the centre of mass was calculated and the maximum intensity was extracted. This is done for each frame individually. Image formation from the 10 frames, which were acquired stepping the illumination over the sample in steps of 500 nm, is done like in a normal scanning transmission microscope. In the latter case, the measured intensity is assigned to the respective position of the raster scan resulting in a two-dimensional map of the intensity transmitted through the sample. In our case, the intensity of each of the multiple foci recorded in the large two-dimensional image is extracted as described above. Each extracted intensity is then assigned to the position of the respective focus just like in the pencil-beam case. Accordingly, in the final reconstructed image every tenth pixel column in the horizontal direction is obtained from one image, while the other nine pixel columns in between originate from the images of the nine other grating positions of the stepping. Owing to a step size of 500 nm for scanning, the horizontal pixel size in the reconstructed image is 500 nm, while the vertical pixel size remains 1.36 μm, i.e. the resulting pixels have a rectangular shape. Therefore, the final image spans the original image size in the vertical direction and 1.36 μm/ (5 μm/10 steps) = 2.72 times the original image size in pixels in the horizontal dimension. The whole procedure described so far is done for the reference scan and the sample scan independently. The last step is the reference correction of the reconstructed sample image dividing it by the reference image. For the analysis, the image was rectified and rotated so that the lines of the resolution test chart are oriented horizontally or vertically, respectively, as the resolution test chart was slightly tilted by 0.9° with respect to the detector. Nearest-neighbour interpolation was performed in this step.

**Modulation transfer function**. The raw data was averaged along the direction of the lines producing the intensity modulation depicted in Fig. 4c. Its mean value was calculated and subtracted. In the next step, we calculated the zero crossings and categorized the intervals into corresponding line pairs in order to calculate their contrast. We averaged the intensity between each zero crossing. The contrast of the line pair is the difference between the corresponding mean values. The calculated MTF is the contrast of a line pair divided by the contrast of the line pair with the lowest spatial frequency. The calculation of the MTF relies on an automatic detection of the maxima and minima of the resolution pattern. As the pattern is crossed by one of the lines of bridges in the grating structure (a sketch of the grating structure is depicted in Supplementary Fig. 4), this automatic detection gets

less reliable towards smaller periods. Thus a manual refinement step was carried out to verify the location of the maxima and minima. However, to have a more robust resolution measure without such a manual refinement step, we decided to determine edge-spread functions in addition, which give us the fundamental resolution limit. The manual refinement of MTF values was thus only carried out to the point where the gain in contrast in the direction of scanning (horizontal direction) could be clearly demonstrated. In Fig. 4d, the discontinuous lines display the MTFs after application of a third-order Savitzky–Golay filter with a width of 21 points. The standard deviation was calculated for the averaging procedure along the direction of the lines of the line pattern and used to calculate the statistical uncertainty of the average. This statistical uncertainty was propagated following the steps of the data analysis described above and the resulting statistical uncertainty is plotted as error bars in Fig. 4d.

**Edge-spread function**. The raw data of the region of interest, indicated by red and blue boxes in Fig. 3a, was averaged parallel to the edge. An error function was fitted to the averaged edge. The sigma value of the error function is the half-period resolution and was compared for both directions in order to identify the gain in resolution.

## Data availability
The data sets generated and analysed during this study are available from the corresponding author upon reasonable request.

## Code availability
The code used for data analysis as well as for display of the data is available from the corresponding author upon reasonable request.

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

## Acknowledgements
Parts of this research were carried out at PETRA III at DESY, a member of the Helmholtz Association (HGF). We would like to thank Helmholtz Zentrum Geesthacht, namely Felix Beckmann, Fabian Wilde and Jörg Hammel for assistance in using P05. We acknowledge financial support through the DFG Gottfried Wilhelm Leibniz program. This work was carried out with the support of the Karlsruhe Nano Micro Facility (KNMF, www.kit.edu/knmf), a Helmholtz Research Infrastructure at Karlsruhe Institute of Technology (KIT).

## Author contributions
B.G. and M.D.: design and realization of experiment, data analysis, manuscript preparation. L.H.: realization of experiment, data analysis, manuscript revision. C.J. and A.H.: realization of experiment, manuscript revision. F.P.: design of experiment, data analysis, manuscript revision.

## Additional information

**Competing interests:** The authors declare no competing interests.

