## [Peer Review File · Nature Communications]

Reviewers' comments:

Reviewer #1 (Remarks to the Author):

The manuscript presents an approach of multi-source STXM to realize the large-area illumination at sub-micron resolution. Instead of using zone-plate optics, the authors utilize the Talbot effect to create a multi-illumination setup, thus speeding up the scan rate significantly. The instrumentation has been demonstrated by both simulation and experiment. The desired resolution is depicted in a 2D image and 1D scans. The method is technically sound, and the novelty is well suited to publish in Nature Communications. Although the time of acquisitions is seemingly much reduced, the authors should clarify this issue more carefully before the manuscript is published.

A fair comparison must be made by calculating the number of photons per image/pixel in different setups. The incident intensity of each pencil beam (of the multi-beam setup) is usually weaker than using a single isolated pencil/parallel beam (i.e., conventional single-source STXM or TXM). It is even much weaker when illumination optics are considered (i.e., the loss due to the grating). The resolution elements per second depend on not only the speed of scan but also the incident flux per unit time. Assuming N illuminations are created at a suitable Talbot distance. The flux of each illumination is $1/N$ of the original incident flux (if the photon loss due to the grating is ignored). Now only one period of scan is needed to obtain the whole image. It is unclear why the quality of the image is superior to the one from the high-speed set-up (i.e., single-source illumination with the same focal spot size but only $1/N$ of the period of scan). The number of photons per unit area per unit time is the same. Similarly, compare to TXM, if the incident flux (before the zone-plate or the grating) is the same, it is also unclear why the total dose can be reduced by a factor of 100 (as addressed in the section of discussion). A larger scan rate does not mean the signal-to-noise ratio of recorded intensities is the same. More specifically, the incident flux right before the specimen must be measured to ensure it is compatible with the conventional STXM and TXM.

It is also notable that the focal spot of STXM can typically achieve 50nm. Even at 20 keV, it can be better than 50 nm most recently. The proposed technique seems more difficult to enhance the probe resolution and only being competitive beyond 30 keV (where the zone-plate optics are much less efficient). Since the achieved resolution is much worse than current STXM imaging, if the aforementioned issue is not fully addressed, the application of this imaging method would be limited to image ultra-thick specimens with sub-micron resolution at high photon energy.

Reviewer #2 (Remarks to the Author):

The authors present a method for obtaining resolution beyond that of a conventional imaging system through the

use of structured x-ray illumination in a parallel beam geometry. The method uses a triangular grating to generate propagated planes with intensity modulations (Talbot carpet) into which a sample (and nearby detector) can be placed. Simulations and measurements of the Talbot carpet are presented along with images of a test pattern which shows higher resolution in the direction orthogonal to the intensity modulations. The analysis procedure is roughly explained.

This work aims to build on the considerable developments of grating interferometry which can provide exquisite phase contrast microscopy with hard x-rays and also provides a simplified experimental arrangement using only a single grating. However I do not see that the results, as presented, support the conclusions. Specifically, the authors do not unequivocally demonstrate that their method improves the resolution of their imaging system. Rather that the imaging system presented has better resolution in one dimension. Second, the description of the analysis routine is very difficult to interpret (seems to leave out critical steps) and doesn't explain the improved bandwidth. For these reasons I cannot recommend publication in this journal. Detailed comments and questions follow below.

1. The majority of the results presented have been done before. The Talbot measurements have been made in reference 14 (at the same facility) where the results seem to be far superior in terms of fringe visibility.
2. The maximum intensity modulation due to the grating is less than 10%. The authors claim there is focusing of the beam due to this modulation but that is not demonstrated. It is not shown whether the maximum intensity in the beam is higher than the incident wavefront. This level of modulation also does not meet the Rayleigh criterion and thus the fringes cannot be considered to be resolved.
3. The authors claim that there is a "gain" in resolution in the horizontal direction due to their method. A gain in resolution is not demonstrated in this paper. A difference between the horizontal and vertical resolutions is presented. Since such differences are common in microscopy it is not possible to judge the value of this technique without seeing conventional images generated without the grating for comparison.
4. Figure 4, the error bars are light grey and almost impossible to see. Why is the apparent spread in the horizontal MTF values so much bigger than the error bars? It seems the error calculation is not adequate and with proper error bars the apparent difference between horizontal and vertical would be lower.
5. In general the description of the experimental arrangement is inconsistent and unclear. The pixel size used for the Talbot carpet measurements was 0.68 microns yet the discussion of the imaging

results (top of page 9) says that the detector pixel size is limited to 1.36 microns. Further, a "resolution" of 0.59 microns is claimed in the horizontal direction. Shannon's sampling theorem would state that obtaining a resolution of 0.59 microns requires a pixel size of at most 0.295 microns yet the authors are claiming to do this with a pixel size that is more than four times bigger. This can't be their intention so the entire discussion of resolution and pixel size is unclear. Furthermore the calculated MTF barely extends beyond 1 micron half-period resolution yet a resolution significantly better is claimed. It is not possible to judge the validity of these claims as currently presented.

6. The method is essentially a shearing interferometer without an analyzer grating, as the authors make 10 measurements at different grating positions across a single grating period. However, it is not discussed at all in the manuscript how the data from the different shearing positions are used or analyzed. The section on signal extraction discusses only a single frame. How are the other frames utilized, why are they measured? It is not discussed.

7. The discussion of pixel size in the "signal extraction" section is again completely unclear. Software binning is used to make an "effective" pixel size of 1.36 microns which is the value they previously stated was the actual pixel size, then later down they claim that in the horizontal direction it is either 2.72 times bigger or smaller, it's not clear. The authors don't seem willing to clearly present the experimental geometry.

8. The signal extraction seems to proceed from a single measurement and appears to actually be some sort of deconvolution, or high pass filter, which may explain the very noisy MTF values. The title of the paper states "structured illumination super-resolution". These words have a specific meaning to microscopists. The analysis requires multiple measurements and relative frequency modulation of the different measurements. Indeed, multiple measurements are made in this case but how they are used is not presented nor is any form of frequency modulation.

Reviewer #3 (Remarks to the Author):

The manuscript presents an innovative method to produce submicron resolution images using fields of view significantly larger than those presently available in microscopy. The subject is of clear importance and the choice of the journal is adequate.

The manuscript reads well and it is clear in all parts.

There are a few points I consider necessary to be addressed (for further clarification and/or for strengthening the conclusions) before publication on NatComm:

1. Include the horizontal and vertical MTF of the detector without gratings, using the same binning mode used with the gratings. This will bring a clear comparison of the gain determined by the new approach, regardless any potential additional effect introduced by the source characteristics.

2. Provide details of the scintillator (material, thickness) because this influences the MTF
3. Please give details on the X-ray source, in particular its size on both directions, and divergence, together with indications (simulations/calculations) on how its characteristics can influence the reported measurements (penumbra)
4. Ideally, I would like to see the experimental MTFs of figure 4d for a gratings placed horizontally.
5. I consider necessary to include a sketch of the gratings structure to allow other teams to potentially reproduce the results.
6. Please correct a typo at line 70, page 3 (latter instead of later).

Point-by-point response to the reviewers' comments

The reviewer comments are formatted in black color while the response is formatted in blue color.

Reviewer #1 (Remarks to the Author):

The manuscript presents an approach of multi-source STXM to realize the large-area illumination at sub-micron resolution. Instead of using zone-plate optics, the authors utilize the Talbot effect to create a multi-illumination setup, thus speeding up the scan rate significantly. The instrumentation has been demonstrated by both simulation and experiment. The desired resolution is depicted in a 2D image and 1D scans. The method is technically sound, and the novelty is well suited to publish in Nature Communications. Although the time of acquisitions is seemingly much reduced, the authors should clarify this issue more carefully before the manuscript is published.

A fair comparison must be made by calculating the number of photons per image/pixel in different setups. The incident intensity of each pencil beam (of the multi-beam setup) is usually weaker than using a single isolated pencil/parallel beam (i.e., conventional single-source STXM or TXM). It is even weaker when illumination optics are considered (i.e., the loss due to the grating). The resolution elements per second depend on not only the speed of scan but also the incident flux per unit time. Assuming N illuminations are created at a suitable Talbot distance. The flux of each illumination is $1/N$ of the original incident flux (if the photon loss due to the grating is ignored). Now only one period of scan is needed to obtain the whole image. It is unclear why the quality of the image is superior to the one from the high-speed set-up (i.e., single-source illumination with the same focal spot size but only $1/N$ of the period of scan). The number of photons per unit area per unit time is the same.

For a pencil beam (STXM-style) scanning at the same resolution, we assume the following parameters: A KB-mirror system with a typical transmission $T_{STXM} = 0.9$ which is capable of focussing the full X-ray beam ($A_{beam} = 5.6\text{mm} \times 3\text{mm}$ in our case) into the same focal spot size ($A_{focus} = 0.7\mu\text{m} \times 0.7\mu\text{m}$) as the gratings. The incoming intensity I_0 is assumed to be the same for the scanning microscope as well as for the structured illumination.

The STXM creates a flux-enhancement

$$FE_{STXM} = \frac{T_{STXM} A_{beam} I_0}{A_{focus}} = 30.86 \cdot 10^6 I_0$$

For the structured illumination, we assume a 2D-grating with $5\mu\text{m}$ period and a detector with an ideal pixel pitch of $5\mu\text{m}$. Including the absorption of the grating, the transmission of the grating into multiple side-by-side foci is $T_{SI} = 0.528$. Instead of the full beam size only the intensity of the area of a single focus ($A_{single-focus} = 5\mu\text{m} \times 5\mu\text{m}$) is focused into A_{focus} . Therefore the flux enhancement for a single focus is

$$FE_{SI} = \frac{T_{SI} A_{single-focus} I_0}{A_{focus}} = 26.94 I_0$$

Accordingly, the acquisition time per point for the scanning approach is reduced to

$$t_{aq,STXM} = \frac{FE_{SI}}{FE_{STXM}} t_{aq,SI} \quad \text{for delivering the same flux at one focus. Therefore the STXM could acquire}$$
$$N_{\text{points,STXM}} = \frac{FE_{STXM}}{FE_{SI}} = 1.15 \cdot 10^6 \text{ points during one acquisition using the structured illumination. In the}$$

latter case, $N_{SI} = \frac{A_{beam}}{A_{focus}}$ are acquired in parallel, where A_{beam} is the area of the beam that is covered by the detector. With $A_{beam} = 5.6\text{mm} \times 3\text{mm}$ in our case, we have $N_{SI} = 672000$. The STXM, without any overhead, could acquire a factor $\frac{N_{STXM}}{N_{SI}} = \frac{T_{STXM}}{T_{SI}} = 1.7$ more points with the same fluence in the same time, which would be in favor of the scanning approach, but only if a KB-mirror system is used. In case that Fresnel zone plates are used, our approach would be faster due to the much lower efficiency of zone plates. Nevertheless, there is one significant drawback for a STXM. With an ideal detector resolution of $5\mu\text{m}$, a camera system with the same chip with 5120×3840 pixels could image a beam of $25.6\text{mm} \times 19.2\text{mm}$. Such beam sizes are readily available at inverse Compton sources and too large for typical KB-mirror systems, thus requiring additional pre-focussing in the

STXM approach.

Furthermore, the speed required for data acquisition in the STXM approach would be extremely high. Our acquisition time was 300 ms per frame, which corresponds to a data acquisition time per illuminated spot of 262 ns in the scanning approach. As a result, the detector has to run at a frame rate of 4 MHz and the sample manipulation stages at a speed of 1910 mm/s during the scan, assuming a continuous scanning mode and neglecting time for de- and re-acceleration after each line. While the envisioned scan ranges are too large for fast piezo actuators, also direct drive stages can achieve this speed (e.g. Standa 8MTL220, Vilnius, Lithuania). However, de- and reacceleration of the stage after each line already takes ~0.2s for the aforementioned stage corresponding to 1200s for the 6000 lines required to scan the whole height of the field-of-view of 3mm with 0.5 μ m spacing. The total scan time would be $t_{STXM} = t_{data-acq.,STXM} + t_{mot.mov.,STXM} = 17.6s + 1200s = 1217.6s$ compared to $t_{SI} = t_{data-acq.,SI} + t_{mot.mov.,SI} = 30s + 2s = 32s$. Consequently, even the STXM employing a KB-mirror system would be slower due to the much larger amount of lines to be scanned compared to our approach, which requires only 10 lines.

We included this discussion as “Supplementary Discussion 1: Performance Calculation” into the supplementary material of the manuscript and referred to this calculation into the manuscript:

“A hypothetical pencil-beam scanning microscope generating the same focal spot size of 0.7 μ m operating at the same X-ray flux suffers significantly from the limited acceleration of state of the art linear stages due to the orders of magnitude larger number of lines to be scanned to acquire the same field-of-view.

As a consequence our method is faster by more than one order of magnitude as demonstrated in the calculation in the Supplementary Discussion 1.”

Similarly, compare to TXM, if the incident flux (before the zone-plate or the grating) is the same, it is also unclear why the total dose can be reduced by a factor of 100 (as addressed in the section of discussion). A larger scan rate does not mean the signal-to-noise ratio of recorded intensities is the same. More specifically, the incident flux right before the specimen must be measured to ensure it is compatible with the conventional STXM and TXM.

In the discussion about the parallel beam microscope, we do not claim that the dose is reduced by a factor of 100, but rather that the dose per image stays the same and that the same amount of images have to be taken in both approaches. The speed-up of our approach is a result of the fact that a higher intensity is impinging the sample as the flux of 5 μ m x 5 μ m is focused and then recorded the detector. 52.8% of this flux is collected in the focus and reaches the detector whereas in the parallel beam microscope no flux enhancement takes place. In order to reach the same spatial resolution, the pixel size has to be 0.5 μ m x 0.5 μ m, which can be reached with an optical microscope magnifying the scintillator plane onto the detector. As the detector area is smaller by a factor 100, the flux enhancement with the structured illumination is 52.8 compared to the parallel beam microscope. Data acquisition with our proposed scanning approach requires 10 x 10 steps resulting roughly in twice the acquisition time as in the conventional parallel beam geometry, but at the same time resulting in 100x the field-of-view of a single parallel beam image. To reach the same field of view, 100 images have to be taken in the parallel beam geometry, too. Therefore, our proposed method is more than 50 times faster than stitching frames from conventional parallel beam imaging with the same resolution, which requires the same number of acquisitions and thus motor movements generating the same amount of data and a similar scanning-overhead. Furthermore, the required optical microscope imaging the scintillator plane onto the detector has typically a low efficiency. Another factor of 2 in efficiency thus could be gained using a fibre-coupled scintillator for the 5 μ m pixels in the structured illumination case, which is not possible for the 10x magnification to 0.5 μ m effective pixel size.

We clarified and detailed the discussion of the parallel beam microscope in the discussion section as described above.

We believe that constructing a classical transmission microscope with a low magnification of 10 is not very useful. Consider a hypothetical TXM with a KB-mirror system ($T_{condensor} = 0.9$) focussing the incoming beam (intensity I_0 , $A_{beam}=5.6\text{mm} \times 3\text{mm}$) down to $A_{focus}=0.56\text{mm} \times 0.3\text{mm}$. In order to achieve the resolution, yet again the beam on the detector has to be $A_{det}=5.6\text{mm} \times 3\text{mm}$. For good

optical image KB-systems are not always the best solution, therefore compound refractive lenses with typical transmissions of $T_{\text{obj.lens}} = 0.4$ might be used as an objective lens.

The intensity in the sample plane is $I_{\text{sam}} = T_{\text{condensor}} I_0 \frac{A_{\text{beam}}}{A_{\text{focus}}}$, the intensity on the detector plane is

$I_{\text{det,TXM}} = T_{\text{obj.lens}} I_{\text{foc}} \frac{A_{\text{focus}}}{A_{\text{det}}} = T_{\text{condensor}} T_{\text{obj.lens}} I_0 \frac{A_{\text{beam}}}{A_{\text{det}}}$. The total scan time is $t = \frac{N_{\text{phot}}}{F_{\text{pixel}}} n_{\text{images}}$, where N_{phot} is the number of photons per pixel required to form an image, $F_{\text{pixel}} = I_{\text{det}} A_{\text{px}}$ is the flux per pixel, A_{px} the detector pixel size and n_{images} is the number of images required to scan the object.

If the same detector is used for both techniques, N_{phot} is the same for both cases, as well as A_{px} . If $\frac{t_{\text{TXM}}}{t_{\text{SI}}} > 1$ our approach is faster. The number of images with our method is 100.

$$\frac{t_{\text{TXM}}}{t_{\text{SI}}} = \frac{\frac{N_{\text{phot}}}{F_{\text{pixel,TXM}}} n_{\text{images,TXM}}}{\frac{N_{\text{phot}}}{F_{\text{pixel,SI}}} n_{\text{images,SI}}} = \frac{F_{\text{pixel,SI}}}{F_{\text{pixel,TXM}} \cdot 100} n_{\text{images,TXM}} = \frac{T_{\text{SI}}}{T_{\text{condensor}} T_{\text{obj.lens}} \frac{A_{\text{beam}}}{A_{\text{det}}} \cdot 100} n_{\text{images,TXM}} > 1$$

For the lenses and grating efficiency discussed before, this results in the condition $n_{\text{images,TXM}} > 68.2$ for our scanning technique to be faster. With $A_{\text{focus}} = 0.56\text{mm} \times 0.3\text{mm}$, the actual amount of scans is $n_{\text{images_req.,TXM}} = \frac{A_{\text{beam}}}{A_{\text{focus}}} = 100$. Accordingly, our method would also outperform such a hypothetical TXM.

We included this discussion in the supplementary material as part of the Supplementary Discussion 1 and referred to it in the manuscript: "Considerations regarding a conventional transmission X-ray microscope are included in this discussion [Supplementary Discussion 1], too."

It is also notable that the focal spot of STXM can typically achieve 50nm. Even at 20 keV, it can be better than 50 nm most recently. The proposed technique seems more difficult to enhance the probe resolution and only being competitive beyond 30 keV (where the zone-plate optics are much less efficient). Since the achieved resolution is much worse than current STXM imaging, if the aforementioned issue is not fully addressed, the application of this imaging method would be limited to image ultra-thick specimens with sub-micron resolution at high photon energy.

We agree with the reviewer's opinion that it might be challenging to push the resolution of this technique below 100 nm. Nevertheless, we believe that our method is a powerful tool to obtain full-field information with sub-micrometre resolution of large specimens ranging up-to several centimetres in size. In such applications, our technique can provide benefits compared to other microscopy techniques as discussed above.

For now, we see our full-field structured-illumination super-resolution X-ray transmission microscopy technique as a complementary technique for larger specimens to the sub-50nm STXM and TXM microscopes which are limited to very small specimen sizes on the order of a few hundred micrometres.

Compared to stitching images in a parallel beam microscope, our approach can achieve a scan time reduction by more than a factor of 50. A combination of our technique with inverse Compton sources paves the way to high-speed sub-micrometre resolution imaging of large specimens in the laboratory. Implementation of a source grating will make this technique feasible also at high-power (rotating-anode) X-ray tubes, significantly decreasing data acquisition time while enlarging the field-of-view compared to current systems usually based on microfocus tubes.

Reviewer #2 (Remarks to the Author):

The authors present a method for obtaining resolution beyond that of a conventional imaging system through the use of structured x-ray illumination in a parallel beam geometry. The method uses a triangular grating to generate propagated planes with intensity modulations (Talbot-carpet) into which a sample (and nearby detector) can be placed. Simulations and measurements of the Talbot-carpet are presented along with images of a test pattern which shows higher resolution in the direction orthogonal to the intensity modulations. The analysis procedure is roughly explained.

This work aims to build on the considerable developments of grating interferometry which can provide exquisite phase contrast microscopy with hard x-rays and also provides a simplified experimental arrangement using only a single grating. However I do not see that the results, as presented, support the conclusions. Specifically, the authors do not unequivocally demonstrate that their method improves the resolution of their imaging system. Rather that the imaging system presented has better resolution in one dimension. Second, the description of the analysis routine is very difficult to interpret (seems to leave out critical steps) and doesn't explain the improved bandwidth. For these reasons I cannot recommend publication in this journal. Detailed comments and questions follow below.

1. The majority of the results presented have been done before. The Talbot measurements have been made in reference 14 (at the same facility) where the results seem to be far superior in terms of fringe visibility.

We disagree with the reviewer on the point that the majority of the results have been done before. The fundamental novelty is our method for super-resolution microscopy exploiting a structured illumination which - to the best of our knowledge - has not been demonstrated before. The Talbot carpet simulation and measurement are presented because we think that it is a necessary characterisation step both in the planning and the conduction of the described experimental procedure. Furthermore, we believe that this measurement together with the simulation contributes to a better understanding of our method, especially for readers who are not familiar with Talbot-carpets or grating interferometry. However, this measurement is not the key result of our manuscript, which is generation of a structured-illumination by a specially designed grating which enables super-resolution microscopy of large specimens, as emphasised in the beginning of the paragraph.

Of course, Talbot-carpet measurements on such a grating have been performed before, which we explicitly pointed out by citing Ref. 14. However, in the case of Ref. 14 this special type of grating was primarily investigated as a potential means to create grating interferometry setups with shorter inter-grating distances, rather than as an approach for super-resolution imaging.

Concerning the far superior fringe visibility in Ref. 14 pointed out by the reviewer, it has to be considered that the pixel size in Ref. 14 was much smaller than in our case, corresponding to a much better intrinsic detector resolution. Accordingly, the Talbot-carpet is not blurred as strongly as in our case by the detector point-spread-function, which is discussed in more detail in the answer to the reviewers comment Nr. 2.

2. The maximum intensity modulation due to the grating is less than 10%. The authors claim there is focusing of the beam due to this modulation but that is not demonstrated. It is not shown whether the maximum intensity in the beam is higher than the incident wavefront. This level of modulation also does not meet the Rayleigh criterion and thus the fringes cannot be considered to be resolved.

The differences between the measured Talbot-carpet and the simulated one can be explained by the detector PSF. This also demonstrates that - although the fringes seem to be smeared out in the measurement - the small foci exist in reality and our assumptions for the scanning approach are valid. We thank the reviewer for his critical comment about amplitude of the measured intensity modulation. We measured 5 frames at each distance and averaged all five of them. Based on the reviewer's comment, we carefully reinspected the raw data frame per frame. It turns out, that at several distances the first two images after motor moves are often heavily distorted, i.e. showing a homogeneous intensity distribution due to residual vibrations after the move. This significantly impaired the modulation depth. Therefore, we used only the last three frames for the analysis now.

This increases the the measured intensity modulation at the focus to 18%-19% including averaging over the bridges which reduces the actual intensity modulation (c.f. Figure 3c in the manuscript). Consequently, the modulation depth fulfills the Rayleigh criterion (~19% modulation for line patterns (Goldstein 1992, <https://doi.org/10.1111/j.1365-2818.1992.tb01517.x>)). In the ideal case (c.f. Figure 1), the intensity modulation would be 100% (from an intensity enhancement of 6 to 0). The difference is mainly a result of the bridges and finite detector point-spread function (PSF) of ~1.5 μm (r.m.s.). Neglecting the detector PSF, the modulation at the focus is ~94% (between ~4.95 at the peak and ~0.3 outside), depicted in Supplementary Figure 1. This is the actual intensity modulation at the specimen. The measured intensity is a convolution of the illumination with the detector PSF. If the measured detector PSF and bridges in the grating structure are included into the simulation (c.f. Supplementary Figure 2), the ideal modulation depth for a fully coherent beam is ~30%. The residual reduction of the modulation is attributed to the unstable X-ray beam and the finite coherence length. In Figure 1. the intensity is normalized to the maximum intensity. We changed the normalization to the incident wavefront in order to demonstrate the ideal flux enhancement resulting in a maximum intensity enhancement by slightly more than a factor of six in the center of the focus. If the detector PSF and bridges of the grating are considered and absorption is neglected, the expected intensity enhancement at the peak measured by the detector decreases from >6 to <1.1. We included this as Supplementary Figure 2 with the same content as Figure 1.

3. The authors claim that there is a "gain" in resolution in the horizontal direction due to their method. A gain in resolution is not demonstrated in this paper. A difference between the horizontal and vertical resolutions is presented. Since such differences are common in microscopy it is not possible to judge the value of this technique without seeing conventional images generated without the grating for comparison.

For a high-resolution X-ray detector optimized for parallel beam imaging, a difference in resolution of a factor of 2.3 between orthogonal directions is not common as these setups are optimized for isotropic resolutions in the image plane (i.e. transverse to the optical axis). In light microscopy, one often sees large differences between lateral and axial resolution, but the in-plane lateral resolution, which is the one relevant for the imaging process of the scintillator onto the camera chip, is usually rather isotropic.

We intended to do the measurement without the grating, but we lost the X-ray beam when switching to a different X-ray energy where we planned to repeat the measurement with the gratings. This was due to a malfunction of the motors moving the monochromator crystals, which could not be recovered during our beam-time. Therefore, the procedure used to focus the visible light objective onto the scintillator is our only measurement of the intrinsic detector resolution, unfortunately.

The intrinsic resolution of the detector system is automatically determined while focussing a 10x visible light objective onto the scintillator. The procedure is based on the PSF calculated from the MTF for knife-edge measurements at different focal positions. At our X-ray energy of 35 keV and for an effective pixel size of 0.68 μm in combination with the 50 μm LuAG-scintillator, the measured PSF is ~5 px (r.m.s.) in focus, corresponding to 3.4 μm . However, this MTF measurement includes diffraction effects caused by the coherence of the beam which lead to an overestimation of the size of the PSF.

Long-term experience of the beamline staff with the same detector configuration has shown that a PSF of ~1.5 μm (r.m.s.) can be reached under these conditions, which matches the vertical PSF determined by our approach. Based on the mentioned long-term experience, the too large value obtained in the MTF measurement also corresponds to an actual PSF of ~1.5 μm (Felix Beckmann, private communication).

4. Figure 4, the error bars are light grey and almost impossible to see. Why is the apparent spread in the horizontal MTF values so much bigger than the error bars? It seems the error calculation is not adequate and with proper error bars the apparent difference between horizontal and vertical would be lower.

We thank the reviewer for pointing out that the error bars in light grey are hard to see on print-outs.

We changed the colour to a darker grey which is much better visible on print-outs.

We included all the statistical uncertainties in the error calculation. Additional fluctuations arising from random vibrations of the monochromator as well as X-ray source position cannot be assessed quantitatively and therefore are not included in the error calculations. Nevertheless, even if only the lowest value of the horizontal MTF, which is at 511lp/mm is considered, it is still much higher than the one in the vertical direction, which decreased already to 0 below 400 lp/mm.

5. In general the description of the experimental arrangement is inconsistent and unclear. The pixel size used for the Talbot-carpet measurements was 0.68 microns yet the discussion of the imaging results (top of page 9) says that the detector pixel size is limited to 1.36 microns. Further, a "resolution" of 0.59 microns is claimed in the horizontal direction. Shannon's sampling theorem would state that obtaining a resolution of 0.59 microns requires a pixel size of at most 0.295 microns yet the authors are claiming to do this with a pixel size that is more than four times bigger. This can't be their intention so the entire discussion of resolution and pixel size is unclear. Furthermore the calculated MTF barely extends beyond 1 micron half-period resolution yet a resolution significantly better is claimed. It is not possible to judge the validity of these claims as currently presented.

At the top of page 9, we write "While the vertical half-period resolution determined with this technique is 1.49 μm which is ultimately limited by the resolution of the detector with a pixel size of 1.36 μm ,...", which does not mean that the detector pixel size is limited to 1.36 μm , but that the half-period resolution cannot be better than the pixel size. The (effective) detector pixel size is dependent on the magnification of the visible light microscope located between scintillator and detector as well as the binning that has been used. For the Talbot-carpet measurement a 10x magnification optic was used creating an effective pixel size of 0.68 μm (and a resolution of $\sim 1.5\mu\text{m}$, c.f. Supplementary Note 1). The same camera system was used for scanning the resolution pattern, but 2x2 software binning resulted in a effective pixel size of 1.36 μm , as described in the methods section (line 222-228 & line 251-253).

Shannon's sampling theorem holds true for a parallel beam microscope, but we are not relying on the detector resolution in our method. Key point of our technique is that the resolution is not limited by the detector, but by the super-resolution achieved by scanning sub-micrometre sized illuminations over the sample. This resolution is independent of the detector pixel size as long as the pixel is not larger than the spacing of the beamlets. Image formation is done like in a normal scanning microscope. The difference is that the measured intensity has to be extracted for the individual illuminations from the large two-dimensional image prior to classical image formation at a scanning microscope, because we are using a structured-illumination instead of a single pencil beam.

Due to a step-size of 500 nm for scanning, the horizontal pixel size in the reconstructed image is 500 nm, while the vertical pixel size remains 1.36 μm , i.e. the resulting pixels have a rectangular shape. Accordingly, every 10th pixel column is obtained from one image, while the other nine pixel columns in between originate from the images of the nine other grating positions of the stepping.

The calculation of the MTF relies on a automatic detection of the maxima and minima of the resolution pattern. As the pattern is crossed by one of the lines of bridges in the grating structure, this automatic detection gets less reliable towards smaller periods. Thus a manual refinement step was carried out to verify the location of the maxima and minima. However, to have a more robust resolution measure without such a manual refinement step, we decided to calculate the edge-spread function in addition, which gives us the fundamental resolution limit. The manual refinement of MTF values was thus only carried out to the point where the gain in contrast in the direction of scanning (horizontal direction) could be clearly demonstrated. For the vertical direction, both resolution values agree very well.

6. The method is essentially a shearing interferometer without an analyzer grating, as the authors make 10 measurements at different grating positions across a single grating period. However, it is not discussed at all in the manuscript how the data from the different shearing positions are used or analyzed. The section on signal extraction discusses only a single frame. How are the other frames utilized, why are they measured? It is not discussed.

Indeed, the set-up looks similar to a shearing interferometer without analyser grating. However, in our case we employ a non-binary phase grating in order to generate a focussing effect resulting in a special Talbot-carpet which creates a structured, periodic and sub-micrometer sized illumination on

the specimen under investigation. Another difference to a classical shearing interferometer is, that we only extract an absorption image in our signal extraction and not a phase image.

In the signal extraction section, we discuss how data from the 10 stepping positions is mapped onto the grid spanned by the total number of points arising from the stepping, i.e. 1 pixel in the vertical direction measures 1.36 μm , while in the horizontal direction 1 pixel measures 500 nm, i.e. the pixels are rectangular with a side-length ratio 1:2.72 (horizontal:vertical). Accordingly the retrieved image has 2.72 times as many pixels in the horizontal direction as in the vertical direction.

However, we acknowledge that the corresponding description in the section on the signal extraction could be improved in order to make it easier to follow the steps of the signal extraction. Therefore, we revised this section significantly.

7. The discussion of pixel size in the "signal extraction" section is again completely unclear. Software binning is used to make an "effective" pixel size of 1.36 microns which is the value they previously stated was the actual pixel size, then later down they claim that in the horizontal direction it is either 2.72 times bigger or smaller, it's not clear. The authors don't seem willing to clearly present the experimental geometry.

We wrote in the main text just pixel size because we thought the specific applied settings (0.68 μm physical pixel size binned to 1.36 μm effective pixel size for the data extraction) were clearly enough explained in the Methods section and therefore thought repeating too much details in the main text might be too confusing. Based on the reviewer's comment, we changed the main text to "(binned) pixel size of 1.36 μm ".

Here, it seems that the signal extraction was not explained clear enough with regard to the image formation. In comment Nr 5 & Nr. 6, we already explained why the amount of pixels in the horizontal direction is 2.72 times larger (due to the stepping with a step-size of 500nm).

We are sorry that our explanation could be confusing for some readers and addressed this issue with a revised signal extraction section.

8. The signal extraction seems to proceed from a single measurement and appears to actually be some sort of deconvolution, or high pass filter, which may explain the very noisy MTF values. The title of the paper states "structured illumination super-resolution". These words have a specific meaning to microscopists. The analysis requires multiple measurements and relative frequency modulation of the different measurements. Indeed, multiple measurements are made in this case but how they are used is not presented nor is any form of frequency modulation.

As explained above in detail, the signal extraction does of course use all 10 acquired measurements. Based on the reviewers comments, we have described the process more clearly in the revised signal extraction section. We do not do any kind of deconvolution or high-pass filtering in our measurement analysis. We analysed our data in real space as it is commonly done in X-ray scanning microscopy, but analysis in frequency-space, which is more common in classical visible light structured illumination microscopy, should yield the same result.

The structured illumination is created by the Talbot-carpet of the triangular grating which creates an array of sub-micrometer wide illuminations which are scanned over one period of the grating across the sample to obtain a full image of the sample. The detector has only to be able to discriminate between individual illuminations, but does not need to fully resolve the intensity profile of the individual illuminations thus creating super-resolution.

Reviewer #3 (Remarks to the Author):

The manuscript presents an innovative method to produce submicron resolution images using fields of view significantly larger than those presently available in microscopy. The subject is of clear importance and the choice of the journal is adequate.

The manuscript reads well and it is clear in all parts.

There are a few points I consider necessary to be addressed (for further clarification and/or for strengthening the conclusions) before publication on NatComm:

1. Include the horizontal and vertical MTF of the detector without gratings, using the same binning mode used with the gratings. This will bring a clear comparison of the gain determined by the new approach, regardless any potential additional effect introduced by the source characteristics.

We intended to do the measurement without the grating, but we lost the X-ray beam when switching to a different X-ray energy where we planned to repeat the measurement with the gratings. This was due to a malfunction of the motors moving the monochromator crystals, which could not be recovered during our beam-time. Therefore, the procedure used to focus the visible light objective onto the scintillator is our only measurement of the intrinsic detector resolution, unfortunately.

The intrinsic resolution of the detector system is automatically determined while focussing a 10x visible light objective onto the scintillator. The procedure is based on the PSF calculated from the MTF for knife-edge measurements at different focal positions. At our X-ray energy of 35 keV and for an effective pixel size of 0.68 μm in combination with the 50 μm LuAG-scintillator, the measured PSF is ~ 5 px (r.m.s.) in focus, corresponding to 3.4 μm . However, this MTF measurement includes diffraction effects caused by the coherence of the beam which lead to an overestimation of the size of the PSF.

Long-term experience of the beamline staff with the same detector configuration has shown that a PSF of ~ 1.5 μm (r.m.s.) can be reached under these conditions, which matches the vertical PSF determined by our approach. Based on the mentioned long-term experience, the too large value obtained in the MTF measurement also corresponds to an actual PSF of ~ 1.5 μm (Felix Beckmann, private communication).

2. Provide details of the scintillator (material, thickness) because this influences the MTF

The scintillator used in the experiment was a lutetium-aluminum-garnet (LuAG) scintillator of 50 μm thickness. We included this information into the following statement in the methods section of the manuscript:

“A 10x optical magnification of the scintillator, lutetium-aluminum-garnet (LuAG) with a thickness of 50 μm , onto the detector resulted in an effective pixel size of 0.68 μm .”

3. Please give details on the X-ray source, in particular its size on both directions, and divergence, together with indications (simulations/calculations) on how its characteristics can influence the reported measurements (penumbra)

In reference nr. 32 (Wilde et al., AIP Conference Proceedings 1741, 030035 (2016)), general parameters of the beam line are presented. The source size (at 10 keV) is specified to be 36 μm x 6.1 μm (hor. x ver.) with a beam divergence of (28 μrad x 4.0 μrad (hor. x ver.)). These parameters change only slightly with X-ray energy. Therefore, we used this horizontal source size for the simulation of source effects (Supplementary Figure 1) as the grating lines were oriented vertically.

Supplementary Figure 1 depicts the effect of the extended source size (and grating bridges) on the Talbot-carpet and the resulting structured illumination. At the position of the experiment, c.f. the vertical cut shown in Supplementary Figure 1c, the source blur barely affects the intensity modulation (source blur $\sim 27\text{nm}$, focus size $\sim 0.7\mu\text{m}$).

We included this statement into the manuscript:

“The effect of the source blur and bridges on the Talbot-carpet are shown in Supplementary Figure 1. At the position of our structured illumination, c.f. Supplementary Figure 1c, the source blur (27 nm) barely affects the intensity modulation (with a width of the individual illuminations of 0.7 μm).”

4. Ideally, I would like to see the experimental MTFs of figure 4d for a gratings placed horizontally.

The horizontal source size is the larger one corresponding to the direction of lower spatial coherence. Orienting the grating lines in the horizontal direction, more triangular structures would interfere compared to the case used in the experiment, where the grating lines were oriented vertically. Accordingly, horizontal gratings should provide stronger interference effects and therefore only improve the results. There is no reason for horizontal gratings to deteriorate the obtained results. Additionally, the current set-up allows only scanning of the grating in the horizontal direction and cannot be converted into one that scans the grating vertically. Therefore, we could not acquire an image with grating lines placed horizontally, but we do not consider this to be necessary, because if our proposed method works in the direction with worse beam properties, which we demonstrated, there is no reason why it should not work in the other direction.

5. I consider necessary to include a sketch of the gratings structure to allow other teams to potentially reproduce the results.

We agree with the reviewer that a sketch of the grating structure can be useful for other teams and included one as Supplementary Figure Nr. 3. We added the following sentence to the grating description in the "Methods" section of the manuscript: "A sketch of the grating structure is depicted in Supplementary Figure 3."

6. Please correct a typo at line 70, page 3 (latter instead of later).

We thank the reviewer for pointing out the typo and corrected it.

Reviewers' comments:

Reviewer #1 (Remarks to the Author):

The authors have carefully addressed the pros and cons of structured illumination by comparing the flux (per unit time) of STXM and TXM. This supplementary information is useful to methodologists and future applications. Although there is in principle no enhancement of flux in K-B mirror system and the probe size is difficult to be further improved as well, in terms of large-area scanning, the setup indeed overcomes the speed limitation of the stage. I am satisfied with the response.

Reviewer #3 (Remarks to the Author):

Authors have addressed reviewers comments by adding new figures (Supplementary materials), reanalyzing data, extending description and comments. All these modifications have been made with care and addressed satisfactorily the concerns raised up by the reviewers.

Only one point (raised up by both Reviewers #2 and #3) remains still open and needs to be solved before I can recommend the publication of the manuscript.

Replies to remarks #1 of Rev #3 and remark #3 of Rev.#2:

Concerning point 1 (request to provide an MTF without gratings), authors gave a perfectly understandable justification on their impossibility to provide the requested plot (technical problems during the slot of time available for data acquisition); however, this justification is not relevant with respect the publication of the manuscript.

Authors estimated an upper limit of the PSF based on a mix of measurements, logic and experience; this methodology is quite slippery. My recommendation to authors is to request the allocation of a new time slot at the laboratory to complete the measurements.

However, if this won't be possible, I would be ready to accept the here presented logic after additional clarifications are given.

a) Authors stated that "long term experience has shown that a PSF of 1.5 microns (r.m.s.) can be reached...". Being the improved resolution one of the key points of the paper it essential that a proof is brought; this proof has to go beyond the "oral communication" even if made by esteemed scientists. I therefore recommend that this statement is supported by data (plot) of previous

measurements made “with the same detector configuration” with a clear description of the detailed parameters used in the measurements.

b) A second concern regards the use of r.m.s. in this context, which sometimes is used as synonym of “standard deviation” and, in more extended meaning, as “sigma” of a Gaussian distribution. R.m.s. intrinsically includes a statistical nature of data, which I cannot see here, unless authors wish to connect it with a not common method used to measure it (to be detailed). Otherwise, authors must disambiguate what they intend with r.m.s. in this context.

c) If authors intend “r.m.s. = standard deviation of a Gaussian distribution” (to be confirmed) then I have additional concerns about its use in this context. One of the classical PSF measurements methods is, in fact, the knife-edge, as also mentioned by authors; in that case the Full width at Half Maximum of the curve is normally considered as the PSF, which, in the Gaussian case is $2.35 \times \text{sigma} = 3.53$ microns and not 1.5 microns as deducted by authors.

Replies to remarks #2, 3,4,5,6 of Rev #3:

Authors have satisfactorily addresses all points, by adding details on scintillator (#2), X-ray source, sketch of the gratings (#5) and having given a clear explanation to question #4.

Additional point: typo on Supplementary information: correct “meters” with “micrometers” in the PSF estimation

Response to Reviewers

Reviewer #1 (Remarks to the Author):

The authors have carefully addressed the pros and cons of structured illumination by comparing the flux (per unit time) of STXM and TXM. This supplementary information is useful to methodologists and future applications. Although there is in principle no enhancement of flux in K-B mirror system and the probe size is difficult to be further improved as well, in terms of large-area scanning, the setup indeed overcomes the speed limitation of the stage. I am satisfied with the response.

Reviewer #3 (Remarks to the Author):

Authors have addressed reviewers comments by adding new figures (Supplementary materials), reanalyzing data, extending description and comments. All these modifications have been made with care and addressed satisfactorily the concerns raised up by the reviewers.

Only one point (raised up by both Reviewers #2 and #3) remains still open and needs to be solved before I can recommend the publication of the manuscript.

Replies to remarks #1 of Rev #3 and remark #3 of Rev.#2:

Concerning point 1 (request to provide an MTF without gratings), authors gave a perfectly understandable justification on their impossibility to provide the requested plot (technical problems during the slot of time available for data acquisition); however, this justification is not relevant with respect the publication of the manuscript.

Authors estimated an upper limit of the PSF based on a mix of measurements, logic and experience; this methodology is quite slippery. My recommendation to authors is to request the allocation of a new time slot at the laboratory to complete the measurements.

We followed the reviewers suggestion and applied for another beam time in order to measure the detector PSF. During this beam time we repeated the measurement of the test pattern with a standard parallel imaging geometry and the same experimental parameters which were used in the experiment with the structured illumination.

Although we used the same 10x magnification optic, the effective pixel size in this case was determined to be 0.642 μm , which is slightly smaller than the 0.68 μm obtained in the first experiment. This might be due to the fact that scintillators are frequently exchanged at the beamline and therefore the scintillator might be at a slightly different position. The same type of scintillator (50 μm LuAG) was used in both experiments. 2x2 binning therefore yields an effective pixel size of 1.284 μm . Data acquisition time was 1000 ms.

We determined the resolution in the vertical and horizontal direction using approximately the same regions of interest like in the structured-illumination case for the edge-measurements.

We acquired dark-frames, reference frames and projections of the test pattern (100 each) and binned all frames 2 by 2 (as done in structured-illumination case). Afterwards, dark frames were subtracted from the projections as well as from the references. Finally a standard reference correction was performed by dividing the projection by the respective reference.

As the X-ray beam was still unstable, which can result in artifacts in the final images (cf. Figure 1 below), reference correction for each projection was first performed separately with each of the

references. Then, the image with the least image deviations in the region of interests for the edge measurement was determined (the procedure is detailed in Supplementary Note 1). We included the reference corrected parallel beam image of the resolution pattern and the edge measurement as Supplementary Figure 1 (the already existing Supplementary Figures were relabelled accordingly).

The resolution determination was performed analogous to the edge analysis in the structured illumination case. The result yields a resolution of $1.45 \mu\text{m}$ (r.m.s) in the vertical direction with the standard parallel beam which is in very good agreement with the $1.49 \mu\text{m}$ (r.m.s.) measured in the vertical direction in the case of structured illumination. In the horizontal direction, the resolution of $1.30 \mu\text{m}$ (r.m.s.) with the parallel beam is slightly better than in the vertical one. Nevertheless, this is still a factor of 2 larger than the resolution in the horizontal direction achieved using structured illumination.

This measurement unambiguously demonstrates that our structured-illumination approach provides a resolution in the horizontal direction beyond detector resolution which is only limited by the size of the individual illuminations.

We believe that this measurement completely addresses the reviewers' last concern about the intrinsic detector resolution.

Figure 1: Two exemplary reference corrected images of the test pattern using the same projection but two different reference frames. **a** unsuited reference frame for this projection, **b** better reference frame for this particular projection.

However, if this won't be possible, I would be ready to accept the here presented logic after additional clarifications are given.

a) Authors stated that "long term experience has shown that a PSF of 1.5 microns (r.m.s.) can be reached...". Being the improved resolution one of the key points of the paper it essential that a proof is brought; this proof has to go beyond the "oral communication" even if made by esteemed scientists. I therefore recommend that this statement is supported by data (plot) of previous measurements made "with the same detector configuration" with a clear description of the detailed parameters used in the measurements.

We addressed this and the following two points by performing a new measurement in standard parallel beam geometry (see response above).

b) A second concern regards the use of r.m.s. in this context, which sometimes is used as synonym of "standard deviation" and, in more extended meaning, as "sigma" of a Gaussian distribution. R.m.s. intrinsically includes a statistical nature of data, which I cannot see here, unless authors wish to connect it with a not common method used to measure it (to be detailed). Otherwise, authors must disambiguate what they intend with r.m.s. in this context.

The meaning of r.ms. depends on the context in which it is used, e.g. pulse lengths or beam radii are commonly given as r.m.s-values in optics, where it is associated with the sigma-value/standard deviation of a Gaussian pulse / Gaussian beam profile shape. We clarified the ambiguity with its use for statistical data by explicitly stating "r.m.s., here and in the following used as the sigma-value of a corresponding Gaussian distribution" where we first mention a width in r.m.s.. Already before, all values in the manuscript were consistently given as these sigma values to ensure comparability.

c) If authors intend "r.m.s. = standard deviation of a Gaussian distribution" (to be confirmed) then I have additional concerns about its use in this context. One of the classical PSF measurements methods is, in fact, the knife-edge, as also mentioned by authors; in that case the Full width at Half Maximum of the curve is normally considered as the PSF, which, in the Gaussian case is $2.35 \times \text{sigma} = 3.53$ microns and not 1.5 microns as deducted by authors.

We regret that our explanation appeared to be unclear. We stated the sigma value of 1.5 μm in order to be consistent with the sigma values determined in the edge-measurements in the experiment, while the same Gaussian intensity distribution (here the PSF) can of course be described in a mathematically equivalent way by the Full Width at Half Maximum value mentioned by the reviewer. As either value describes the same Gaussian curve, we opted to give sigma values (and explicitly marked them as such) as they are used consistently throughout the manuscript and are also direct fit parameters here.

Replies to remarks #2, 3,4,5,6 of Rev #3:

Authors have satisfactorily addresses all points, by adding details on scintillator (#2), X-ray source, sketch of the gratings (#5) and having given a clear explanation to question #4.

Additional point: typo on Supplementary information: correct “meters” with “micrometers” in the PSF estimation

We thank the reviewer for noticing the typo. As this section has been reworked completely, the sentence containing this typo is no longer included in the supplementary information.

REVIEWERS' COMMENTS:

Reviewer #3 (Remarks to the Author):

I fully acknowledge the work made to address my concerns and in particular the realization of an additional experiment, which has now permitted to give more solid bases to the results. I am fully satisfied of the way the answer has been composed; however, I have to remark a few minor points (Supplementary Note 1).

- Resolutions are given with a different number of significant digits (see Figure 1 caption, for instance): 1.3 microns in the horizontal direction, and 1.45 in the vertical one. I believe that 1.3 is a typo, (missing 0) but it must be corrected
- Along the same lines, Supplementary Note 1, I read "the effective pixel size in this measurement was 0.642 μm ": it is hard to believe that measurements had a few nm resolution to justify the use of 4 significant digits at this stage; which was the associated error? In other words, this value needs to be re-evaluated by authors.

All other points (in particular those related to the r.m.s.) have been addressed and clarified.

Response to Reviewer

Reviewer #3:

I fully acknowledge the work made to address my concerns and in particular the realization of an additional experiment, which has now permitted to give more solid bases to the results. I am fully satisfied of the way the answer has been composed; however, I have to remark a few minor points (Supplementary Note 1).

- Resolutions are given with a different number of significant digits (see Figure 1 caption, for instance): 1.3 microns in the horizontal direction, and 1.45 in the vertical one. I believe that 1.3 is a typo, (missing 0) but it must be corrected

We thank the reviewer for pointing out the typo. We corrected the value given in the caption from 1.3 to 1.30.

- Along the same lines, Supplementary Note 1, I read “the effective pixel size in this measurement was 0.642 μm ”: it is hard to believe that measurements had a few nm resolution to justify the use of 4 significant digits at this stage; which was the associated error? In other words, this value needs to be re-evaluated by authors.

The effective pixel size was determined from the pixel size and the measured magnification. Focussing the visible light objective and determination of the magnification is done with a beamline script which does not provide the error associated with the measurement. The value given in the Supplementary Note 1 was calculated from the magnification determination performed directly before the scan. An error of 1% in the measurement of the magnification would already change the effective pixel size by a few nm. Therefore, we agree with the reviewer on the matter that 4 significant digits are too accurate and we rounded the effective pixel size to 3 significant digits in the final version of the Supplementary Information.

All other points (in particular those related to the r.m.s.) have been addressed and clarified.